# An Online-Based Transition Care Program for Adolescents with Spina Bifida Using Intervention Mapping: A Protocol for Program Development

**DOI:** 10.3390/ijerph19031056

**Published:** 2022-01-18

**Authors:** Eun Kyoung Choi, Hyeseon Yun, Eunjeong Bae

**Affiliations:** 1Mo-Im Kim Nursing Research Institute, College of Nursing, Yonsei University, Seoul 03722, Korea; ekchoi@yuhs.ac; 2Brain Korea 21 FOUR Project, College of Nursing, Yonsei University, Seoul 03722, Korea; firstmind47@gmail.com

**Keywords:** spina bifida, adolescents, transition care, online program, intervention mapping

## Abstract

Adolescents with spina bifida (SB) face challenges in their transition to adulthood due to intensive medical regimens and delayed development of independence. Despite an increasing interest in the transition of adolescents with SB to adulthood, the clinical evidence of transition care remains limited, and existing studies have focused on the effects of intervention programs. This study aims to describe the process of systematically developing an online-based transition care program for adolescents with SB using the intervention mapping (IM) protocol. IM consists of six steps: (1) logic model of the problem; (2) program objectives; (3) program design; (4) program production; (5) plan to implement the program; (6) plan for evaluation. At first, five problems faced during the transition were identified, based on which four program objectives and six program strategies were established. The online transition care program for adolescents with SB was developed as a six-week program. The main strength of this program is that it reflects the diverse perspectives of adults with SB and health care professionals and is easy to apply because it is online. We aim to further validate the feasibility of this transitional care program to evaluate its effect based on our evaluation plan.

## 1. Introduction

Spina bifida (SB) is one of the most common congenital malformations, known as neural tube defects, caused by a failure in the closure of the vertebrae during the early stage of pregnancy [1]. Before 1960, the survival rate for patients with SB was about 10–12%; however, owing to recent advancements in health care, over 85% of the children with SB are expected to now survive to adulthood [2]. As SB is a common congenital malformation with high survival until adulthood, it must now be approached as a chronic condition [3,4]. Therefore, it is of great importance to improve life prospects for adolescents with SB and ensure that they live healthy and independent [3,5].

From a health care perspective, the transition is the purposeful, planned passage of adolescents with chronic conditions from child-centered to adult-oriented health care systems [6]. Adolescence is a challenging time of physical, psychological, and social change [7]. Adolescents with chronic conditions face even greater challenges because they also have to deal with important changes in the care they need and how it is provided [7]. Moreover, Health care transition services that fail to adequately meet the needs of young adults with chronic conditions at this time of considerable change may result in a deterioration in health status that can have negative long-term consequences [7].

Adolescents with SB have experienced challenges in their transition to adulthood due to physical and psychological difficulties [3]. Those with SB often have hydrocephalus with a shunt, bladder and bowel dysfunction, urinary and fecal incontinence, orthopedic deformities, motor difficulties, and pressure sores [1,6,8]. Because these physical difficulties constitute an important trigger of psychological difficulties [9,10], adolescents with SB also experience psychosocial difficulties such as depression, a decreased quality of life, anxiety, and social immaturity [11,12,13]. Additionally, adolescents with SB also encounter challenges in achieving their adulthood milestones, including education, employment, and romantic relationships [14].

Individuals with SB need to manage their complex treatment regimen throughout their lives, which includes regular hospital visits, clean intermittent catheterization (CIC), bowel management, skin checks, shunt monitoring, and the use of various orthopedic devices [2,15]. Adolescence is a key developmental stage during one’s course of life [16]. In particular, the responsibility of care is shifted from parents to young adolescents with chronic conditions; however, in the case of SB, the shift to autonomy is delayed, thereby increasing the challenges faced during the transition from adolescence [5,17]. Unfortunately, adolescents with SB are less likely to be ready for the transition than adolescents with or without other chronic conditions such as type 1 diabetes and Turner syndrome [18]. If the patients do not smoothly transition to adulthood, inadequate self-care can lead to increased hospitalization and serious complications, such as kidney transplantation [19,20]. Due to the increasing interest in the transition of adolescents with chronic conditions to adulthood, research on the development and evaluation of transition programs for SB has been actively conducted since 2010 [21,22,23,24]. Despite an established consensus about the importance and need for the transition care program for adolescents with SB, clinical evidence for care during this transition remains limited [25]. Transition care programs should support adolescents in their broader transition to adulthood, including personal, social education, and vocational needs, not just their medical needs [26]. However, most transition care programs focus on disease management [22,25,26]. Adolescents with SB need to develop goals related to health care and for potential higher educational and vocational plans with other long-term plans [27]; therefore, an integrated transition management program that includes these aspects is needed. Moreover, all existing studies have focused on the effects of the interventions, and it is difficult to find studies that systematically present the process of program development [21,22,23,24]. In other words, there is currently a lack of information on the development process, content, and structure of transition care programs, including comprehensive content for SB. To address these concerns, we provided a detailed methodological description of the process of developing the transition care program using the intervention mapping (IM) method.

IM is a systematic process for developing theory-, evidence-, and practice-based programs [28]. It aims to provide guidelines for effective decision-making during the development of an intervention by integrating theory, empirical findings, and information from the target population [28]. Therefore, developing the transition care program for adolescents with SB according to the IM method might provide guidelines based on empirical evidence and theory to clinical practice. To further facilitate these predicted positive results, we planned to engage with experts (multidisciplinary panel) who have rich clinical practice experience in the program development. IM has been applied extensively to develop complex promotion programs for health-related behavior [28,29,30,31]. More recently, several previous studies have suggested that IM offers a process to develop an intervention focused on the health of children and adolescents [28,29,31]. In summary, if the process of developing a transition care program is presented systematically and in detail, it could enable health care professionals to use that information as fundamental data to apply tailored transition care programs based on the environment and characteristics of various clinical settings and communities.

Among the theoretical frameworks explaining the transition of adolescents with SB, the Life Course Model is considered as providing a useful basis [3,16,32]. The Life Course Model was laid out developmentally, from preschool and continuing through school-age, and adolescence and young adults. This model provides specific guidance on how to prepare for the transition and manage them in each developmental stage. This model also consists of three domains: self-management/health (self-care), personal and social relationships, and education/employment [16,32]. Based on this model, we planned to develop programs considering the three domains that should be carried out in adolescence [32]. Furthermore, in the Life Course Model, as the family factor also influences the transition [32], it was considered to involve parents in the program.

Considering the COVID-19 pandemic, developing an online-based transition care program for adolescents with SB would be especially useful. Therefore, this study aimed to describe the process of systematically developing an online-based transition care program for adolescents with SB using the IM protocol.

## 2. Materials and Methods

IM consists of six steps, and each step is built on the decisions and results obtained from the preceding step [33]. The six steps are as follows: (1) logic model of the problem; (2) program objectives (logic model of change); (3) program design; (4) program production; (5) plan to implement the program; and (6) plan for evaluation. We recruited a multidisciplinary panel of four young adults with SB (19–25 years), two pediatric urologists, two pediatric nurse practitioners, and a pediatric orthopedist to develop the online-based transition care program.

### 2.1. Step 1: Logic Model of the Problem

To identify the problems related to transition, we conducted a systematic review [25], a need assessment [3], and a discussion with the multidisciplinary panel. First, for conducting a systematic review of transition programs for adolescents with SB [25], we performed a search on PubMed, Web of Science, CINAHL, and PsycINFO databases for studies published from January 2010 to June 2019. The search terms were (“adolescent*” or “youth” or “young adult*” or “teen*” or “adolescent” or “young adult”) AND (“spina bifida” or “myelomeningocele” or “spinal dysraphism” or “myelomeningocele” or “spinal dysraphism”) AND (“transition*” or “transfer*” or “continuity of care” or “continuity of patient care”). After excluding the studies according to the criteria, i.e., peer-reviewed journals and English language, we ultimately analyzed eight out of 218 studies. Additionally, additional literature reviews were conducted as required by the research team.

Second, we conducted a cross-sectional study [3] to explore the educational need of adolescents with SB from January 2020 to January 2021. Finally, a total of 108 adolescents and young adults with SB aged 13–25 years were included in the final analysis. The research team developed the questionnaires to evaluate the educational needs of young adults with SB transitioning to adulthood based on previous studies on the assessment of transition needs of these individuals [2,34,35]. The questionnaires consisted of six domains: the system of care (four items), the psychological context for self-management (six items), neurosurgery (two items), orthopedics and physical activity (two items), urology and sexual health (six items), and specific health issues including bowel management, skin damage prevention, puberty, and nutrition (six items). Lastly, the needs and contents of the transition care program were outlined through discussions with the multidisciplinary panel. The results of the systematic review and needs assessment are presented in other articles [3,25].

### 2.2. Step 2: Program Objectives (Logic Model of Change)

We organized the objectives of the program to indicate which behaviors needed to be changed or achieved. After selecting the problems related to the transition in the first step, we conducted an additional literature review for establishing the program objectives. Finally, in step 2, the program objectives and rough contents of the transition care program were specified.

### 2.3. Step 3: Program Design

The program was conceptualized and designed in this step. The research team established the theoretical framework and components of the program. We integrated the information acquired from previous steps, i.e., the literature review and researchers’ discussions, and used it to construct the transition educational program. This theoretical framework guided us to better understand the problems faced during the transition, to optimize factors that promote transition readiness, and to suggest solutions and outcomes.

### 2.4. Step 4: Program Production

In this step, we determined the details of the topics, scope order, and sequence of the transition care program. Performance and implementation strategies were selected based on the discussion among the research team to ensure more effective delivery methods.

### 2.5. Step 5: Plan to Implement the Program

We developed a plan to apply specific methods in the program and discussed the barriers and facilitators of the program. In regard to the educational aspect of this program, we obtained expert advice from three pediatric nurse practitioners and two social workers. The researcher team then worked with the experts to create lecture materials based on the contents of the program. We used animated content that described the transition of adolescents with SB from adolescence to adulthood to increase the participants’ understanding of this transition and to generate interest in the program.

### 2.6. Step 6: Plan for Evaluation

We developed an evaluation plan by finalizing the study design, main concepts, and methods for evaluating the program’s effectiveness. A group of experts conducted the content validity index (CVI) for evaluating the program’s validity. A checklist was developed to assess the level of transition for adolescents with SB, and the CVI was assessed by the multidisciplinary panel. In addition, the template of the individual transition plan was modified based on previous research [36] to establish individual transition goals.

## 3. Results

### 3.1. Step 1: Logic Model of the Problem (Needs Assessment)

Based on the systematic review [25] and need assessment [3] on the transition program, the needs were identified for an intervention that aims to support the transition process among adolescents with SB. Through the systematic review on transition programs for adolescents with SB [25], we found that the characteristics of participants and intervention strategies were the main components of the transition care program. Therefore, we decided to establish an individual transition plan by taking into account the clinical and general characteristics of the participants.

From a needs assessment study [3], adolescents with SB required further education in a few areas, namely “health insurance system”, “SB related urinary system diseases management”, “SB related nervous system symptoms”, and “self-catheterization management.” Additionally, in a previous study, “catheterization”, “bowel program”, “skin care”, “medication”, “exercise”, and “setting up health care appointments” were the tasks that had to be transferred from the parents to the adolescents with SB during the transition period [5]. Following this, discussions with the multidisciplinary panel were conducted based on the literature review results and the needs assessment of transition care programs for adolescents with SB. Finally, five general problems affecting the transition of adolescents with SB were: (1) health care issues (understanding SB, medication, bowel management, urological health, neurological health, orthopedic health, and sexual health); (2) medical treatment-related issues (health insurance system and management of health care-related appointments); (3) daily and school life; (4) relationship with family members; and (5) preparing for the future (career) (Table 1). Through Step 1 of IM, the broad program goal was set to improve the transition readiness for adolescents with SB.

### 3.2. Step 2: Program Objectives (Logic Model of Change)

At first, the research team created a mind map to visualize the goals to be achieved during the transition by referring to a previous study [37]. The goals were as follows: (1) taking care of their own health; (2) getting along well with family and friends; and (3) finding the right school and jobs (Appendix A). Based on the mind map and step 1 of IM, we received advice from the multidisciplinary panel to make the program objectives more concrete. Finally, four program objectives were selected (Table 1), which were as follows: (1) establishing the initiative in self-health care; (2) redefining relationships with friends and family; (3) preparing for independence within the family; (4) exploring educational courses and occupations. Finally, six program strategies were established to achieve the program objectives. The six program strategies were (Table 1): (1) assessing the participants’ characteristics and transition to establish the transition plans; (2) setting the individual transition plans according to the characteristics and transition level of participants through online counseling for each family; (3) providing information about self-health care through online education program taught by health care experts; (4) sharing their worries and issues caused by SB and finding solutions through online discussion; (5) mentoring program to motivate adolescents with SB for successful transition and future; (6) involving parents in family counseling to sharing their children’s transition levels and set and evaluate transition plans together.

### 3.3. Step 3: Program Design

We selected the Life Course Model for SB [32] as the framework of the transition care program. The Life Course Model suggests that it is necessary to prepare for the transition during the developmental stage in the major life areas of self-management/health (self-care), personal and social relationships, and education/employment (major life areas). Moreover, family, environmental, and personal factors can influence the transition process. In particular, the Life Course Model strongly emphasizes the need for “taking the lead on managing primary and secondary conditions”, “friendship and independence within the family”, and “exploring education and career options for meaningful occupation” for the transition of adolescents with SB. According to the Life Course Model, we hypothesized that the transition care program would lead to a successful transition by affecting the domains of self-management/health (self-health care initiative), personal and social relationships (redefining relationships with friends and family), preparing for independence within the family, and education/employment (exploring educational and occupational opportunities) (Figure 1).

In the Life Course Model, self-management includes discrete functional domains related to bladder and bowel management, mobility, skin integrity, sexuality [16,32]. Therefore, health care professionals will give health education to improve self-management knowledge, including these contents, and an animation about the transition will be provided to adolescents with SB to help better understand the transition concept. Since the Life Course Model emphasizes goal setting for self-management and the role of the family in the transition process [16], self-management goals will be established and evaluated through counseling with parents. The personal and social relationship domains include the personal development of adolescents with SB, as well as their relationship with family, friends, and romantic partners [16]. For personal and social relationships, adolescents with SB will share sincere stories about personal relationships and give solutions to each other through group discussion. Lastly, the Life Course Model suggests a mentoring program for adolescents with SB by adults with SB. Thus, we planned a mentoring program about education courses and occupational opportunities with adults with SB.

### 3.4. Step 4: Program Production

The online transition care program for adolescents with SB was developed as a six-week program (Table 2). The program will be conducted every week with real-time participation on Zoom, and each session will be 60–100 min long. The intervention will involve multiple teaching methods (including a lecture and discussion), as well as role-playing and group activities (sharing experiences and finding a solution).

In the first session, adolescents and their parents (caregivers) will be individually interviewed, and the transition program will be introduced through animation. Following this, the degree of transition of each participant will be assessed, and the individual transition plans and goals will be established through discussion with the participants and their parents. Sessions 2–5 would consist of group activities. In the second session, lectures on neurological health, including the general understanding of SB, and orthopedics health, including awareness on skincare, will be provided. After that, we aim to provide time for questions and answers related to the topic and hold group counseling for cognitive behavioral motivation (CBM) related to relationships with family members. After a lecture on voiding and bowel management in SB, a question-and-answer session will be conducted in the third session. Then, group counseling on CBM related to relationships with friends will be conducted. In the fourth session, sexuality education specific to the general population and to those with SB will be conducted, and CBM group counseling will be held, focused on school experiences, and potential solutions will be discussed. In the fifth session, a lecture on the health care system will be conducted, including making hospital appointments and insurance-related information. Next, we will invite an adult mentor with SB to share their experiences at school (relationship with others) and to provide career-related tips. The sixth session will proceed similar to the first session. After evaluating the degree of transition after the program, the degree of implementation of the individual transition plans and goals will be evaluated. We also aim to discuss long-term transition goals within the family.

### 3.5. Step 5: Plan to Implement the Program

We plan to recruit participants by collaborating with the largest SB clinic in South Korea and the Korean Spina Bifida Patients Association. For adolescents with SB, the eligibility criteria would include the following: (1) adolescents with SB aged 12–15 years; (2) ability to use the internet; and (3) absence of cognitive impairment. Adolescents who (1) had already participated in the transition program, (2) were unable to communicate and respond to the questionnaires, or (3) had illness or disabilities other than SB will be excluded. To increase the understanding of transition among adolescents with SB, we created an animation introducing this transition process. With the help of the animation director, we created a storyboard on the transition care of adolescents with SB. Eventually, we created three-minute Korean (https://youtu.be/Ti0YX1trtXI (accessed on 13 January 2022)). and English versions (https://youtu.be/AXUr8WaEXyA (accessed on 13 January 2022)). of the animation that provide an overview of the transition process. This animation will be shown to the participants at the beginning of the program. Educational videos on neurological health, orthopedics health, the hospital system, and sexual issues faced by adolescents with SB will be filmed, featuring experts (three pediatric nurse practitioners) and the research team. We will make the participants watch these educational videos together while other programs are being conducted in real-time.

### 3.6. Step 6: Plan for Evaluation

According to the program objectives set through Steps 1–2 of the IM process, the hypotheses will test whether the transition readiness improved in the intervention group relative to the comparison group.

The CVI of the educational content of the transition care program was evaluated by experts. The group of experts comprised two pediatric urologists, a pediatric orthopedist, two child health nursing professors, and a pediatric nurse practitioner, all of whom had either directly cared for patients with SB or were experts in SB. The result of overall CVI was 0.99, and the item content validity index (I-CVI) ranged from 0.83 to 1.00. Moreover, the program contents were supplemented according to experts’ opinions that easy terms and pictures are needed to help adolescents with SB understand self-management. For program performance, the research team, three pediatric nurse practitioners, and two social workers wrote a program scenario and discussed how to proceed with the program. The research team pre-practice was also conducted, such as confirming the use of content sharing and group division for smooth use of Zoom in the transition program.

To measure the effectiveness of the program, we will measure the appropriate concepts based on the framework of the program; self-management/health domain (transition readiness and sexual health), personal and social relationships domain (social support, including relationships with friends, teachers, and family), and education/employment domain (career readiness). The first and sixth sessions will be conducted online with the participants and their families to establish and evaluate the individual transition plans. For the individual transition plans, we will use the template provided in a previous study on the transition of adolescents with SB [36]. We will amend and use the template of the individual transition plan for the Korean context. This template consists of prioritized goals, current status and plans, actions for planning, target dates, and complete dates (Appendix A). To establish the individual transition plans, we have developed a checklist to assess the current transition levels of adolescents with SB (Appendix A). The checklist comprises 50 questions distributed among five domains, based on previous studies [2,3,36,37,38], and five general problems affecting the transition of adolescents with SB, established based on our results. The CVI of the checklist was assessed by the multidisciplinary panel. The result of overall CVI was 0.96, and the I-CVI ranged from 0.78 to 1.00.

To confirm the effects of the developed program, we would use a quasi-experimental pre-post design. According to the rule stating a sample size of 12 per group for a pilot study [39], we will recruit 12 participants per group. The data of participants will be collected using self-reported questionnaire surveys. The assessment time points will be before the program, immediately after program completion, and four weeks after program completion. The data will be analyzed using IBM SPSS Statistics software version 26 (IBM Corp., Armonk, NY, USA). The Cronbach’s alpha will be measured for the internal consistency of the scales. Descriptive statistics will be used to summarize the characteristics of the participants. For the assessment of sample homogeneity, Fisher’s exact test and the χ2 test will be used. To determine the effect of the pre-post program among the groups, repeated measures ANOVA will be conducted.

## 4. Discussion

This study described the systematic process of developing an online-based transition care program for adolescents with SB. Although most previous studies have been conducted with a focus on the effects of interventions during the transition of this population [21,22,23,24], there is insufficient evidence on the effectiveness of these transition programs [7], and the components of previous interventions are unclear as most of them have not included the target population during program development [40]. Therefore, the present study is meaningful because it is the first attempt to systematically present the process of developing a program for the transition of adolescents with SB to adulthood. IM used in this study is a systematic process that emphasizes theory-, evidence-, and practice-based intervention [33]. We, therefore, used IM to develop a theory-, evidence-, and practice-based program for adolescents with SB, despite it being a time- and effort-consuming process.

The first strength of our program is that it emphasizes the practical problems faced during transition through a literature review [25] and assesses the needs of the transition program for adolescents with SB [3]. Based on this, we aim to establish the individual transition plans considering each participant’s clinical and general characteristics. Moreover, we identified that adolescents with SB have high educational requirements during the transition period to understand the health insurance system, SB-related urinary disease management, SB-related nervous system symptoms, and self-catheterization management. These findings enabled us to design our program with integrated individual and group activity sessions. Therefore, this program will provide content that is tailored (i.e., individual transition plan) and general (i.e., education and group activities) for the successful transition of adolescents with SB.

The second strength of the program is that it would lead to parental involvement in the transition care program. Family is an important factor during the transition of adolescents with chronic conditions [41], and this program will help to increase the parents’ interest and awareness of their child’s transition by encouraging participation while establishing transition plans.

The third strength of the program is that it includes a multidisciplinary panel that consists of four young adults with SB (19–25 years), two pediatric urologists, two pediatric nurse practitioners, and a pediatric orthopedist. The program content accuracy can be verified with the participants of a large number of experts who have the latest information on how to educate individuals with SB. Moreover, we conducted the study through discussions and obtained advice from the panel at each step of the IM. The contents of the program reflect the experience of transitioning participants with SB and the clinical practice of health care professionals.

The final strength of the program is that it is an online-based transition care program, which makes it easily accessible by the participants and their families [42]. Furthermore, being able to participate regardless of the geographical location is the most significant advantage of online programs. Adolescents living in distant regions from hospitals usually face challenges in accessing these health education programs; therefore, this program could be a good opportunity for them to be involved. Moreover, an online-based program is also more useful during the current COVID-19 pandemic. Above all, the online-based program can allow more anonymity than face-to-face programs [42]; therefore, it could encourage sincere involvement from the participants who are socially anxious and reserved.

### 4.1. Implication for Transition Care Programs

This study suggests important implications of transition care for adolescents with chronic conditions.

First, health care professionals should establish tailored transition plans by monitoring the clinical characteristics and transition levels of adolescents with SB. Moreover, health care professionals should make an effort to raise awareness and interest in transition among children and adolescents by explaining the concept of transition from childhood. Second, health care professionals should provide transition care programs that parents can participate in with their children. As parents participate in the program together, the transition care will be managed at home, which will promote the transition of adolescents with SB. Third, Adolescents with SB should be provided with an integrated transition program that focuses on physical self-management and includes interpersonal relationships, education courses, and occupational opportunities.

Therefore, health care professionals should provide counseling that considers these aspects and, if necessary, link with the local community. Government-level support and efforts are also needed to create an environment and policies that can provide an integrated transition program to youth with chronic diseases.

### 4.2. Limitations and Recommendations

This study has some limitations. First, we developed the transition checklist based on the literature review and expert advice to assess the current level of transition for adolescents with SB; however, the checklist is limited due to a lack of validation. Therefore, it is unclear whether this checklist accurately captures the transition levels in adolescents with SB. In the future, we recommend validating this checklist in adolescents with SB for the accurate identification of transition levels. Second, although this study followed IM, a systematic intervention development process, we could not confirm its effectiveness. Therefore, we aim to conduct further studies that evaluate and demonstrate the effectiveness of the transition care program for adolescents with SB. Third, when comparing online programs with face-to-face programs, there is a limit to non-verbal communication because of the screen limitation of the online interface. Moreover, passive participation may occur depending on the participants’ characteristics, or they may feel difficult to interact. Therefore, the health care professionals and the researchers who conducted the transition care program should try to promote participation by providing questions and feedback to the participants so that continuous interaction can occur. For this, it will be necessary to pre-practice the online program in advance and understand the characteristics of the subject. Lastly, it can be difficult to collaborate with experts who have experience caring for SB in clinical practice. Therefore, it is expected that continuous multidisciplinary research and practice will be carried out in the future, an expert network will be formed, and eventually, various studies on the transition of chronic conditions will proceed.

## 5. Conclusions

A systematic approach for developing a transitional care program is necessary to ensure the successful transition of adolescents with SB to adulthood. The transition care program, developed with a systematic approach, could provide fundamental data for developing an intervention program to guide the transition of adolescents with SB. The feasibility of this transitional care program will be validated to evaluate the effect of this program based on our evaluation plan.

## Figures and Tables

**Figure 1 ijerph-19-01056-f001:**
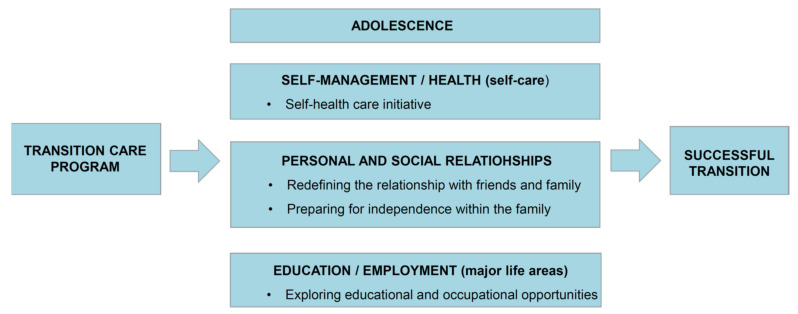
The framework of the transition care program.

**Table 1 ijerph-19-01056-t001:** The problems of transition, program objectives, and program strategies.

Problems of Transition	Program Objectives	Program Strategies
Health care issues - Understanding SB - Medication - Bowel management - Urological health - Neurological health - Orthopedic health - Sexual health Medical treatment-related issues - Health insurance system - Management of health care-related appointment Daily and school life Relationship with family members Preparing for the future (career)	Establishing the initiative in self-health care Redefining relationship with friends and family Preparing for independence within the family Exploring educational courses and occupations	Assessing the participants’ characteristics and level of transition Setting the individual transition plans Providing information about self-health care Sharing their worries caused by SB Mentoring program to motivate for the successful transition and future Increasing family involvement in the program

SB = spina bifida.

**Table 2 ijerph-19-01056-t002:** The contents of the transition care program.

Session (Time)	Contents	Life Course Model Domain	Method
I (60 min) Individual	Overview of transition Introduce the program	Self-management/Health - Self-health care initiatives	Animation ^a^
	Individual transition plans and goals	Personal and social relationship - Preparing for independence within the family	Discussion ^a^
II (100 min) Group	Understanding of SB Walking and skin management	Self-management/Health - Self-health care initiatives	Lecture and discussion
	CBM group counseling - Relationship with family members	Personal and social relationship - Redefining the relationship with friends and family	Group activity ^b^
III (90 min) Group	Voiding management Bowel management	Self-management/Health - Self-health care initiatives	Lecture and discussion
	CBM group counseling - Relationship with friends	Personal and social relationship - Redefining the relationship with friends and family	Group Activity ^b^
IV (90 min) Group	Sexual health	Self-management/Health - Self-health care initiatives	Lecture and discussion
	CBM group counseling - School life	Personal and social relationship - Redefining the relationship with friends and family	Group Activity ^b^
V (100 min) Group	Health care system	Self-management/Health - Self-health care initiatives	Lecture and discussion
	Mentoring program	Education/Employment - Exploring educational and occupational opportunities	Group Activity ^c^
VI (40 min) Individual	Evaluation of individual transition plans and goals	Personal and social relationship - Preparing for independence within the family	Discussion ^a^

CBM = cognitive-behavioral motivation; SB = spina bifida; ^a^ with parent; ^b^ adolescents with SB-led; ^c^ with mentor.

## Data Availability

Data sharing not applicable to this article as no datasets were generated or analyzed during the current study.

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
