# Peer review of "An Online-Based Transition Care Program for Adolescents with Spina Bifida Using Intervention Mapping: A Protocol for Program Development"

_ijerph, 2022, doi:10.3390/ijerph19031056_

Round 1
Reviewer 1 Report
This is a very important area of research considering the difficulties young people with SB encounter as they transition across services and the lack of resources available currently misaligned with their needs. Transition models in place do not respond efficiently to pre-existing and ongoing challenges and limited service provision intensifies the current gap.
Lines 32-35: Quite repetitive therefore and again therefore is used. It is of great important/vital to improve life prospects for adolescents with SB and ensure that they live healthy and independent lives.
Lines 36-37: This definition of transitions is pertinent to healthcare/institutional transitions although several different transitions such as developmental, biological, cognitive are taking place at the same time. This overlap should be considered.
Adolescence is a critical stage and risk period for the emergence of mental health disorders and BF constitutes an important risk factor/trigger. On top of that, institutional transitions can exacerbate ongoing mental health problems. Some discussion around these interaction effects amongst these variables should be included.
The theoretical and empirical frameworks in the introduction should be stronger along with the rationale for the present approach/strategy to develop the proposed program. Relevant theories of development could be embedded to inform this section and the present program development. What theories have informed the current model? Any therapeutic models/modalities to inform the module content? Further empirical studies in this area need to discussed. There are references to certain intervention studies although their content is not outlined along with their limitations and the need for the current IM approach.
It looks as this program was coproduced and codesigned with experts by experience. There’s no specific mention in the introduction about this process and what are the benefits of involving members with lived experience in research. This is a very important area to discuss and illustrate this as a major strength of the proposed intervention.
Some background on the perplexities of complex interventions and the difficulties encountered during implementation stage(s) should be added to justify the current approach and how this model will lead to gradual and effective implementation that is aligned with healthcare providers training, education, and experience.
What about heterogeneity and diversity amongst young people with diverse needs e.g., those with BF and mental health problems and/or other physical/cognitive impairments? This will jeopardise the fidelity of the feasibility study. How will you account for this?
Lines 62-65: Is it the healthcare providers who are to develop transitional care programs? They need to use evidence-based programs which have been tested out with the use of feasibility studies to check if they can be practical and acceptable in clinical settings although they need to collaborate with academic health scientists and coproduce such care models combining empirical with clinical experience.
Line 65 It is unclear what you mean by developmental study here. Do you mean a gradually developed study/model outlining the processes/stages involved in producing an online-based transition program?
Based on Table 1 it is not quite clear how the program objectives will be achieved and they need to be delineated in more depth.
The logic model should be part of the evaluation plan.
The program design needs to go beyond descriptive information and titles of the course modules. How will these be achieved practically? For the program production section, what is the rationale of some sessions being individual-and group-based?
Lines 268-278 A feasibility study is not the same as an evaluation which is the next step. The plan for evaluation so possibly running a fully powered trial should follow the feasibility trial. The feasibility study needs to check the acceptability, practicality, adherence, fidelity etc. of the proposed intervention so basically whether this online-based intervention can be applied in this group and healthcare providers would accept its delivery. The plan for evaluation section needs to be updated and follow feasibility study guidelines. Implementation would follow evaluation- a program to be rolled out and implemented needs to be evaluated first in terms of its effectiveness and cost-effectiveness.
Parental involvement is presented in the discussion as an important theme/element for the program development although the introduction does not discuss lack of parental involvement as a protective factor to optimal transitions and positive/successful transition outcomes.
Lines 324-333: How will the limitations be addressed, and potential risks mitigated? Any plans in action? What about the mode of delivery being online? Any associated limitations versus to face-to-face? What are the challenges you encountered when involving experts by experience?
What are the implications for policy and clinical practice? These should be considered and mentioned as a separate section in the discussion.
Reviewer 2 Report
Thank you for the opportunity to review this technical paper about the development of your intervention for young people with Spina Bifida in transition to adulthood.
The manuscript clearly outlines how this intervention is a response to a previously reported area of need in this population. The methodological steps are clearly described using the 6 steps of Intervention Mapping. The content of the programme is responsive and the delivery methods reflect the current move towards digital innovations.
I think the paper will be of interest to many services supporting young poeple in transition to adulthood.
In my view, this manuscript is written well and I recommend it to the editor
Reviewer 3 Report
Dear authors
I would like to congratulate you on your educational initiative. It is necessary to address the needs of our patients, especially during these hard pandemic times. Consequently, your work in online programs is much interesting.
There are a lot of strong points to identify in your manuscript. The introduction is well-grounded, and the plan is clear and explained.
Strengths and limitations are fully described.
Please check the following items before doing anything else:
Videos are not available to watch (lines 243,244)
Round 2
Reviewer 1 Report
All the suggested comments have been addressed and the manuscript has improved substantially.
Author Response
We are grateful for this insightful comment on the paper.
